# Associations between Blood Metabolic Profile at 7 Years Old and Eating Disorders in Adolescence: Findings from the Avon Longitudinal Study of Parents and Children

**DOI:** 10.3390/metabo9090191

**Published:** 2019-09-19

**Authors:** Diana L. Santos Ferreira, Christopher Hübel, Moritz Herle, Mohamed Abdulkadir, Ruth J. F. Loos, Rachel Bryant-Waugh, Cynthia M. Bulik, Bianca L. De Stavola, Deborah A. Lawlor, Nadia Micali

**Affiliations:** 1Medical Research Council Integrative Epidemiology Unit at the University of Bristol, Bristol BS8 2BN, UK; diana.santosferreira@bristol.ac.uk (D.L.S.F.); d.a.lawlor@bristol.ac.uk (D.A.L.); 2Population Health Sciences, Bristol Medical School, University of Bristol, Bristol BS8 2PS, UK; 3Social, Genetic & Developmental Psychiatry Centre, Institute of Psychiatry, Psychology & Neuroscience, King’s College London, London SE5 8AF, UK; christopher.huebel@kcl.ac.uk; 4UK National Institute for Health Research (NIHR) Biomedical Research Centre, South London and Maudsley Hospital, London SE5 8AF, UK; 5Department of Medical Epidemiology and Biostatistics, Karolinska Institutet, SE-171 77 Stockholm, Sweden; cynthia_bulik@med.unc.edu; 6University College London, Great Ormond Street Institute of Child Health, London WC1N 1EH, UK; moritz.herle.12@ucl.ac.uk (M.H.); r.bryant-waugh@ucl.ac.uk (R.B.-W.); b.destavola@ucl.ac.uk (B.L.D.S.); 7Department of Psychiatry, Faculty of Medicine, University of Geneva, CH–1205 Geneva, Switzerland; mohamed.abdulkadir@unige.ch; 8Icahn Mount Sinai School of Medicine, New York, NY 10029, USA; ruth.loos@mssm.edu; 9Department of Psychiatry, University of North Carolina at Chapel Hill, Chapel Hill, NC 27599, USA; 10Department of Nutrition, University of North Carolina at Chapel Hill, Chapel Hill, NC 27599, USA; 11Bristol National Institute of Health Research Biomedical Research Centre, Bristol BS1 3NU, UK; 12Child and Adolescent Psychiatry Division, Department of Child and Adolescent Health, Geneva University Hospital, CH–1205 Geneva, Switzerland

**Keywords:** metabolomics, nuclear magnetic resonance, EDTA-plasma, eating disorders, anorexia nervosa, binge-eating disorder, ALSPAC

## Abstract

Eating disorders are severe illnesses characterized by both psychiatric and metabolic factors. We explored the prospective role of metabolic risk in eating disorders in a UK cohort (*n* = 2929 participants), measuring 158 metabolic traits in non-fasting EDTA-plasma by nuclear magnetic resonance. We associated metabolic markers at 7 years (exposure) with risk for anorexia nervosa and binge-eating disorder (outcomes) at 14, 16, and 18 years using logistic regression adjusted for maternal education, child’s sex, age, body mass index, and calorie intake at 7 years. Elevated very low-density lipoproteins, triglycerides, apolipoprotein-B/A, and monounsaturated fatty acids ratio were associated with lower odds of anorexia nervosa at age 18, while elevated high-density lipoproteins, docosahexaenoic acid and polyunsaturated fatty acids ratio, and fatty acid unsaturation were associated with higher risk for anorexia nervosa at 18 years. Elevated linoleic acid and n-6 fatty acid ratios were associated with lower odds of binge-eating disorder at 16 years, while elevated saturated fatty acid ratio was associated with higher odds of binge-eating disorder. Most associations had large confidence intervals and showed, for anorexia nervosa, different directions across time points. Overall, our results show some evidence for a role of metabolic factors in eating disorders development in adolescence.

## 1. Introduction

Eating disorders commonly emerge in adolescence or young adulthood and can have a relapsing course with severe long-term consequences including comorbid psychiatric disorders, such as depression and obsessive-compulsive disorder, as well as increased mortality and suicide [1,2]. All eating disorders are characterized by disordered eating behaviors, ranging from extreme restriction, commonly seen in anorexia nervosa, to uncontrolled binge eating reported by patients with binge-eating disorder, alongside cognitive symptoms such as extreme focus on and dissatisfaction with weight and shape [3]. Eating disorders have complex etiologies, influenced by genetic, behavioral, and environmental factors [4]. Therefore, the application of metabolomics in eating disorders may help disentangle the contributions of these components. The metabolome is the furthest downstream product of genome–environment interactions, potentially rendering it a direct and sensitive measure of eating disorders phenotypes at the molecular level. Large-scale genome-wide association studies (GWAS) of anorexia nervosa have interrogated the metabolic component of this illness [5,6]. Results indicated a positive genetic correlation between genetic variants associated with anorexia nervosa and high-density lipoprotein cholesterol, as well as negative genetic correlations with fasting insulin and fasting glucose [7]. The role of insulin in anorexia nervosa has been described previously [7,8]. A systematic review [7] highlighted that anorexia nervosa is associated with increased insulin sensitivity whilst bulimia nervosa and binge-eating disorder are associated with decreased insulin sensitivity. Furthermore, longitudinal research has indicated that starvation, common in anorexia nervosa, can have long-term metabolic consequences [8]. Even though eating disorders can affect people across the entire weight spectrum, binge-eating disorder and overweight/obesity commonly co-occur [9,10]. Indeed, previous research has highlighted that binge-eating disorder is associated with a high burden of metabolic risk factors and particularly higher fasting glucose (the latter independently of body mass index (BMI)) [10].

It is often assumed that metabolic differences are consequences of eating disorders; however, previous research has highlighted that eating disorders are preceded by differential BMI trajectories as well as persistent eating perturbations, such as childhood fussy eating, overeating, and fasting [11,12,13]. These findings suggest the possibility that individuals at risk for eating disorders show differential metabolic profiles prior to diagnosis. However, it is unknown if differences in metabolic markers precede the onset of eating disorders. In this exploratory study, we assessed for the first time the prospective associations between blood metabolic markers at 7 years and anorexia nervosa and binge-eating disorder in adolescence, using data from the Avon Longitudinal Study of Parents and Children (ALSPAC), a large UK birth cohort. Logistic regression was used to examine the associations of each EDTA-plasma metabolic traits at 7 years old with either of anorexia nervosa and binge-eating disorder assessed at 14, 16, 18, and cumulatively by 18 years.

## 2. Results

We investigated the association between 158 metabolic traits at 7 years with anorexia nervosa (Figure 1 and Figure 2) and binge-eating disorder (Figure 3 and Figure 4) at 14 (blue), 16 (red), 18 years (green), and cumulatively until 18 years (black). For our main analyses, we fitted 1264 logistic regression models that indicate whether an elevated metabolic trait level is associated with lower or higher risk of an eating disorder diagnosis in adolescence. We present the data by plotting the point estimates (circles or dots) and respective confidence intervals (error bars) in forest plots (Figure 1, Figure 2, Figure 3 and Figure 4) where metabolic traits are clustered by biochemical classes defined by core chemical structure (e.g., fatty acids) or metabolic pathway (e.g., glycolysis). In forest plots, the points to right of the vertical black line that marks the baseline risk (i.e., 1) indicate elevated risk for an eating disorder whereas points on the left of the black vertical line indicate lower risk for an eating disorder in adolescence.

### 2.1. Descriptives

Table 1 shows the distributions of participant characteristics at 7 years (age at metabolic profiling) included in this study (*n* = 2929) and stratified by eating disorder diagnosis at age 14, 16, 18 years old and cumulatively across all available time points (from 14 until 18 years). We only included those ALSPAC participants who had available metabolomic and phenotype data.

In our subsample, 4%, 3%, 2%, and 10% of participants met diagnositic criteria for anorexia nervosa at 14, 16, 18 years, and cumulatively at all three time points, respectively (Table 1). Binge-eating disorder was diagnosed in 1%, 1%, and 2% at 14, 16, and 18 years, respectively, and in 3% of participants by age 18 years (Table 1). Adolescents who experienced anorexia nervosa or binge-eating disorder at any of the time points were primarily female.

On average, adolescents who experienced anorexia nervosa at any of the time points had lower BMI and lower total calorie intake at age 7 compared with individuals who did not experience an eating disorder in adolescence (all *p*-value < 0.006). Only average calorie intake at 7 years for participants who experienced anorexia nervosa at 18 years was comparable with controls (*p*-value = 0.23). At age 7 years and on average, participants who later developed binge-eating disorder had comparable BMIs and total calorie intake to individuals free from eating disorders at any later time point. Only participants with binge-eating disorder at age 16 had higher BMIs than the reference group at age 7 years (*p*-value = 0.005). Maternal education was comparable between participants diagnosed with either anorexia nervosa or binge-eating disorder and participants without any eating disorders.

In addition, Appendix A presents characteristics for the following: all offspring in ALSPAC (*n* = 19,290), offspring that attended 7 years clinic (*n* = 8293) and offspring included in this study (*n* = 2929). Participants with information on any of the exposures of interest (*n* = 2929) had higher educated mothers when compared to all offspring in ALSPAC (50% vs. 23% with mothers with more than 12 years in education, respectively).

### 2.2. Metabolic Markers Associated with Anorexia Nervosa

First, we present that lipid markers elevated at age 7 years were associated with lower risk for anorexia nervosa at age 18 years, indicated by an odds ratio (OR) below 1. Higher concentrations of very low-density lipoprotein (VLDL) (all subclasses except very small VLDL), very low-density lipoprotein particle size (OR = 0.5, 95% CI = 0.3 to 0.9), total triglycerides (OR = 0.4, 95% CI = 0.3 to 1), very low-density lipoprotein triglycerides (for example, extremely large, OR = 0.4, 95% CI = 0.2 to 0.9), and monounsaturated fatty acid ratio (OR = 0.6, 95% CI = 0.4 to 0.9) at 7 years were associated with lower odds for anorexia nervosa at age 18 years (green filled points in Figure 1 and Figure 2 and Appendix A).

Second, the following lipid markers at age 7 years were associated with higher risk for anorexia nervosa at age 18 years, indicated by an OR above 1: Elevated high-density lipoprotein particle size (OR = 1.6, 95% CI = 1.1 to 2.3), cholesterol in high-density lipoprotein (for example, total cholesterol in large high-density lipoprotein, OR = 1.6, 95% CI = 1.1 to 2.4), fatty acid unsaturation (OR = 1.6, 95% CI = 1.1 to 2.4), docosahexaenoic acid ratio (OR = 1.4, 95% CI = 1.1 to 1.9) and polyunsaturated fatty acid ratio (OR = 1.7, 95% CI = 1 to 2.6) at age 7 years were associated with higher odds for anorexia nervosa at age 18 years (green filled points in Figure 1 and Figure 2, Appendix A).

Higher apolipoprotein B/A-I at 7 years old was associated with lower odds of anorexia nervosa at 14 (OR = 0.8, 95% CI = 0.6 to 1, blue point Figure 1) and 18 years (OR = 0.6, 95% CI = 0.4 to 1, green point Figure 1). No glycolysis-related trait nor amino acids measured at age 7 years showed an association in our exploratory study.

### 2.3. Metabolic Markers Associated with Binge-Eating Disorder

First, we present the lipid markers that when elevated at age 7 years were associated with lower risk for binge-eating disorder at age 16 years, indicated by an odds ratio (OR) below 1: Free cholesterol (OR = 0.6, 95% CI = 0.4 to 0.97), total cholesterol (OR = 0.6, 95% CI = 0.4 to 1.0), linoleic acid (OR = 0.6, 95% CI = 0.3 to 0.9), polyunsaturated fatty acid ratio (OR = 0.7, 95% CI = 0.5 to 0.97), and acetate (OR = 0.5, 95% CI = 0.3 to 1.0) were associated with lower odds of binge-eating disorder at 16 years (red filled points in Figure 3 and Figure 4, Appendix A). In addition, linoleic acid ratio and n-6 fatty acid ratio were also associated with lower risk for binge-eating disorder at 16 years and with binge-eating disorder cumulatively by 18 years.

Second, the following lipid markers at age 7 years were associated with higher risk for binge-eating disorder at age 16 years, indicated by an OR above 1: Higher saturated fatty acid ratio at 7 years was associated with increased odds of binge-eating disorder at 16 years (OR = 1.6, 95% CI = 1.0 to 2.5). We observed no association between glycolysis-related metabolic traits and amino acids at age 7 years and binge-eating disorder in adolescence.

### 2.4. Sensitivity Analyses

We also conducted a sensitivity analysis (see Section 4.4.) to explore how much of the association was confounded by BMI or calorie intake. Findings were broadly comparable across models with similar point estimates and overlapping CI (for example, for the association between anorexia nervosa and total fatty acids at 18 years, all point estimates were approximately OR = 0.8, 95% CI = 0.5 to 1.2). For anorexia nervosa, associations for creatinine attenuated by BMI adjustment across the four time points. Forest plots comparing across models with different adjustments can be found in Appendix A.

## 3. Discussion

To the best of our knowledge, this is the first study to explore the prospective role of metabolic risk factors in the development of eating disorders. In this population-based study of adolescents who experienced anorexia nervosa or binge-eating disorder at ages of 14, 16, 18 or lifetime up to 18 years, we detected alterations in lipid traits at age 7 years that were associated with higher or lower odds of anorexia nervosa at 18 years or binge-eating disorder at age 16 years. We did not detect alterations in glycolysis-related metabolic traits or amino acid concentrations at age 7 that were prospectively associated with an eating disorder diagnosis.

In contrast to our study, previous research on blood markers and eating disorders has primarily focused on anorexia nervosa and explored the hypothesis that metabolic changes are a consequence of eating disorders by comparing individuals with current anorexia nervosa with individuals after weight recovery and/or healthy controls. These approaches did not address whether metabolic alterations are detectable in individuals who go on to develop an eating disorder later in life.

In addition, only one study [8] has used a metabolomics platform (Biocrates Life Sciences, mass spectroscopy, 163 metabolites, 29 women, 16 years old) to quantify metabolic traits. The remainder used standard biochemistry assays. The problem with biochemistry assays is that they only measure an extremely limited number of metabolic traits in human specimens, such as blood, cerebrospinal fluid, sputum, or feces. To overcome this limitation, we measured detailed blood metabolic traits (*n* = 158 traits). Commonly, metabolomics platforms quantify hundreds of metabolic traits from a biological sample (two orders of magnitude more than clinical chemistry assays) providing a broader molecular signature from multiple metabolic pathways potentially affected by eating disorders and therefore offer greater coverage than standard biochemistry assays.

Due to the limited number of previous studies and differences in methods, direct comparison between our study and previous literature is difficult as we explored the hypothesis that metabolic abnormalities are ‘traits’ of eating disorders and therefore precede their onset and may be classified as risk factors.

In our analysis, the following lipid traits were associated with lower odds for anorexia nervosa at age 18 years: We observed lower odds of anorexia nervosa at age 18 years with elevated very low-density lipoprotein at age 7 years independent from BMI. Very low-density lipoproteins transport endogenous produced lipids from the liver to peripheral tissues. A meta-analysis observed no association between very low-density lipoprotein and anorexia nervosa in acutely ill patients [14]. However, only three studies with 90 anorexia nervosa cases were included to obtain the pooled estimate and heterogeneity was high. Additionally, measurements in weight-recovered anorexia nervosa patients were missing.

A recent GWAS reported a positive genetic correlation between anorexia nervosa and high-density lipoprotein cholesterol [5], meaning that patients with anorexia nervosa carry genetic variants that predispose them to elevated high-density lipoprotein cholesterol concentrations. High-density lipoproteins transport cholesterol from peripheral tissues to the liver. Our results were in line with this observation, we found higher odds of anorexia nervosa at 18 years with higher concentration of high-density lipoproteins at age 7 years.

Furthermore, we observed an association between elevated very low-density lipoprotein triglycerides at age 7 years with lower risk for anorexia nervosa in adolescence. No differences in very low-density lipoprotein triglycerides have been reported in an observational study comparing acutely ill anorexia nervosa patients with healthy controls [15]. Additionally, a higher concentration of triglycerides [14] has been observed in anorexia nervosa patients; however, they tend to normalize after weight restoration [5]. In contrast, we found that higher concentrations of triglycerides at 7 years were associated with lower risk of anorexia nervosa at 18 years.

Hypercholesterolemia is a common finding in acutely ill anorexia nervosa patients with higher levels of total cholesterol. However, total cholesterol of anorexia nervosa patients returns to concentrations within the normative range after weight recovery [5]. We did not observe a prospective association of total cholesterol with anorexia nervosa in adolescence, suggesting that hypercholesterolemia may be a state rather than a trait in anorexia nervosa.

Moreover, both GWAS [5,6] reported a negative genetic correlation between anorexia nervosa and fasting glucose, suggesting that patients with anorexia nervosa carry genetic variants that predispose them to lower fasting glucose concentrations. We, however, found no associations between anorexia nervosa case status in adolescence and glucose or glycolysis-related metabolic traits in childhood. However, this finding must be interpreted extremely cautiously as non-fasting blood samples were taken at age 7.

Currently, the literature regarding biochemistry in binge-eating disorder is very limited and no GWAS of binge-eating disorder has yet been performed that would allow the interpretation of potential shared genetics between metabolic traits and binge-eating disorder. However, our study has generated new hypotheses regarding the potential implication of certain metabolic traits in the development of binge-eating disorder. We observed an association between higher cholesterol and higher linoleic acid at age 7 years and lower risk for binge-eating disorder at age 16 years. Linoleic acid is an essential fatty acid that must be obtained through diet. We, therefore, hypothesize that children who develop binge-eating disorder later in life may show altered eating behavior as early as age 7 years. In fact, previous research in the sample has suggested that children who were reported to persistently overeat during the first ten years of life were at higher risk of engaging in binge eating in adolescence [12].

Previous genetic, retrospective case-control, and prospective research showed that a high BMI is associated with higher risk of binge-eating disorder [16,17,18,19,20] and metabolic and weight abnormalities seen in binge-eating disorder are for the most part opposite to those seen in anorexia nervosa [21]. Nonetheless, in our study, individuals who experienced binge-eating disorder in adolescence had, on average, comparable BMIs to their peers at age 7 years; except those who experienced binge-eating disorder at age 16 who had higher BMIs than the reference group at age 7 years (*p*-value = 0.005). Participants who experienced anorexia nervosa in adolescence had, on average, lower BMIs at 7 years old than the reference group.

Overall, we did not observe differences in direction of associations between metabolic traits and binge-eating disorder across time points (Appendix A). It is, however, unclear, why we observed differences for most metabolic traits in the directions of associations with anorexia nervosa across time points. It is possible that at each time point the anorexia nervosa group reflects different etiologies of eating disorders with early onset cases capturing more severe cases. Alternatively, it could be an artifact of the very small sample size and small number of cases. Additionally, anorexia nervosa diagnosis at 18 years did not include parental report, unlike previous time points. This heterogeneity in direction of associations across the three time points may explain why point estimates for anorexia nervosa in the cumulative analysis, across time points until 18 years old, were mostly null.

In models without BMI adjustment, higher creatinine at 7 years was associated with a lower odds of anorexia nervosa in adolescence. After adjusting for BMI, associations attenuated towards the null, suggesting that body size drove the association between creatinine and anorexia nervosa as total muscle mass, which is captured by BMI, is the most important determinant of the creatine pool size and therefore of creatine production.

Limitations should be considered when evaluating the results of our study. First, blood samples were drawn from non-fasting participants. If individuals who experience eating disorders in adolescence exhibited eating disorder behaviors as early as age 7, then blood fasting status could confound associations for fasting-labile metabolic traits. However, most blood metabolic traits seem to be unaffected by fasting status [22,23,24,25,26]. Even so, glucose and extremely large very low-density lipoprotein traits (as defined by this NMR platform, see Appendix B) may be affected as the latter have possible contributions of chylomicrons which are particles, only produced after meals, that transport dietary lipids from the intestines to other locations in the body. Nonetheless, we anticipate that most of this potential confounding would be captured by BMI and calorie intake measured at 7 years that we included in sensitivity analysis due to their association with eating behaviors. However, our results must be seen in the light of this limitation and future studies and data collection must sample participants after an appropriate fasting period.

Second, we cannot exclude the possibility of selection bias due to loss-to-follow-up. We found that participants included in this study had more educated mothers compared to all offspring-participants included in ALSPAC.

Third, due to the small number of individuals who experienced eating disorders and resulting uncertainty of our effect estimates (particularly for binge-eating disorder), these findings need to be replicated in larger cohorts. However, currently no other cohort that assessed metabolic profiles and eating disorder pathology prospectively is available. Hence, this is an exploratory study, and the first to assess the prospective role of metabolic risk factors in eating disorders development in which we focused on general patterns of associations.

Fourth, currently no single or combination of metabolomics technologies can characterize the entire metabolome [27]. Nonetheless, the metabolic traits measured represent a broad molecular signature of systemic metabolism and many of which are related to multiple pathways affected in eating disorders, such as lipids, glycolysis-related metabolites, and amino acids, which exceeds any previous study of biomarkers in eating disorders.

In conclusion, we have shown evidence of some association between metabolic factors in childhood and development of eating disorders in adolescence. We observed differences in very low-density lipoprotein, high-density lipoprotein, and fatty acid traits in anorexia nervosa, and total and free cholesterol and fatty acid traits in binge-eating disorder. Our findings encourage replication and extension in larger well-characterized cohorts as such data become available.

## 4. Materials and Methods

### 4.1. Study Population

We studied offspring participating in the ALSPAC, a prospective pregnancy/birth cohort that was established to explore how genetic and environmental characteristics influence health and development in parents and children [28,29]. During 1990-92, recruitment enrolled pregnant women resident in the former region of Avon, South West of England, with an expected date of delivery between 1st April 1991 and 31st December 1992. Offspring who were alive at 1 year (*n* = 13,988) have since been followed at regular intervals [29], with an additional 713 children enrolled over the course of the study.

Ethical approval for the study was obtained from the ALSPAC Ethics and Law Committee and the Local Research Ethics Committee and all ethical codes can be found on http://www.bristol.ac.uk/medialibrary/sites/alspac/documents/governance/Research%20Ethics%20Committee%20approval%20references.pdf. Ethical codes for 7 year old clinic are United Bristol Healthcare Trust: E4168 ALSPAC Hands on Assessments at Age Seven (30th September 1998); Southmead Health Services: 67/98 Avon Longitudinal Study of Pregnancy and Childhood (ALSPAC) Hands on Assessments at Age Seven (14th September 1998); Frenchay Healthcare Trust: 98/52 Avon Longitudinal Study of Pregnancy and Childhood (ALSPAC); Hands on Assessments at Age Seven (8th December 1998). Consent for biological samples has been collected in accordance with the Human Tissue Act (2004). Informed consent for the use of data collected via questionnaires and clinics was obtained from participants following the recommendations of the ALSPAC Ethics and Law Committee at the time. The main caregiver initially provided consent for child participation and from the age 16 years the offspring themselves have provided informed written consent. Details of all data are available through a fully searchable data dictionary at www.bristol.ac.uk/alspac/researchers/our-data.

The current study included 127 and 36 offspring who had experienced anorexia nervosa or binge-eating disorder, respectively, by age 18 and up to 2240 controls from whom a blood metabolome profile at 7 years of age was available.

### 4.2. Eating Disorders Diagnoses and Covariates

Anorexia nervosa and binge-eating disorder diagnoses, at 14, 16, and 18 years, according to DSM-5 criteria, were obtained as described previously [20]. Anorexia nervosa or binge-eating disorder by age 18 was defined as all individuals who were diagnosed with anorexia nervosa or binge-eating disorder at any one of the three time points: 14, 16, or 18 years, and had not crossed over to another eating disorder. The reference group was individuals who had not experienced any eating disorder, nor eating disorders symptoms by age 18 years, as well as participants who had not experienced any eating disorder, nor eating disorders symptoms, at 14 and 16 years, but had missing data at 18 years. Maternal education was obtained by questionnaire during pregnancy and used as an indicator of family socioeconomic position (A levels or higher, lower than A levels; A levels are needed to enroll in university in the UK). The participants’ sex was obtained from obstetric records and age was calculated from their dates of birth and dates of questionnaires or clinic assessments at age 7 years. BMI was calculated from measurements collected during clinical assessments when participants were approximately 7 years old. Height was measured in light clothing without shoes to the nearest 0.1 cm using a Harpenden stadiometer and weight was recorded to the nearest 0.1 kg using a Tanita scale. BMI was calculated as weight (kg) divided by the square of height (m^2^). Calorie intake at 7 years was assessed through a food frequency questionnaire as extensively described previously [30].

### 4.3. EDTA-Plasma Metabolome Profiling

A high-throughput (^1^H) NMR metabolomics platform, Nightingale Health^©^ (Helsinki, Finland), was used to quantify 158 metabolic traits in children’s non-fasting EDTA-plasma collected at 7 years old. The platform quantifies routine lipids, 14 lipoprotein subclasses, including particle concentration and lipids transported by these particles (see Appendix B), various fatty acids and fatty acids traits (e.g., chain length, degree of unsaturation, see Appendix B), amino acids, ketone bodies, glycolysis and gluconeogenesis-related metabolites, fluid balance, and one inflammation-related metabolite. This set of metabolic traits represent a broad molecular signature of systemic metabolism [31] and most are quantified in clinically meaningful concentrations (e.g., mmol/L). Fatty acids were modelled in original units and as ratios (expressed in %) to total fatty acids. Details of this platform and its use in epidemiological studies have been described elsewhere [32,33,34,35,36]. In addition, we calculated the phenylalanine/tyrosine ratio as it was shown to be a marker of catabolism in individuals with anorexia nervosa [37,38].

### 4.4. Statistical Analysis

All metabolic traits were standardized by subtracting the sample mean and dividing by the sample standard deviation (SD), as this allows comparison across metabolic traits with different units and/or different concentration ranges. Logistic regression was used to examine the associations of each EDTA-plasma metabolic traits at 7 years old with either anorexia nervosa or binge-eating disorder assessed at 14, 16, 18, and cumulatively by 18 years. Associations are reported as odds ratios (OR) reflecting the change in odds of anorexia nervosa or binge-eating disorder per 1 SD increase in metabolic trait concentration at 7 years old. We fitted four models: in model 1, ORs were adjusted for maternal education and child’s sex and age; model 2, had the same covariates as model 1 plus child’s BMI at 7 years; model 3, was as model 1 plus child’s calorie intake at 7 years; and model 4 as model 1 with further adjustment for child’s BMI and calorie intake at 7 years. These confounders were decided a priori, we report the results of model 4 in our main analyses and comparison across models is shown in Appendix A. Our study is exploratory and we focus on effect size and precision [39,40]. All analyses were conducted using Stata version 14.1 (Stata Inc., TX, USA) and R version 3.5.1 (R Foundation for Statistical Computing, Vienna, Austria).

## Figures and Tables

**Figure 1 metabolites-09-00191-f001:**
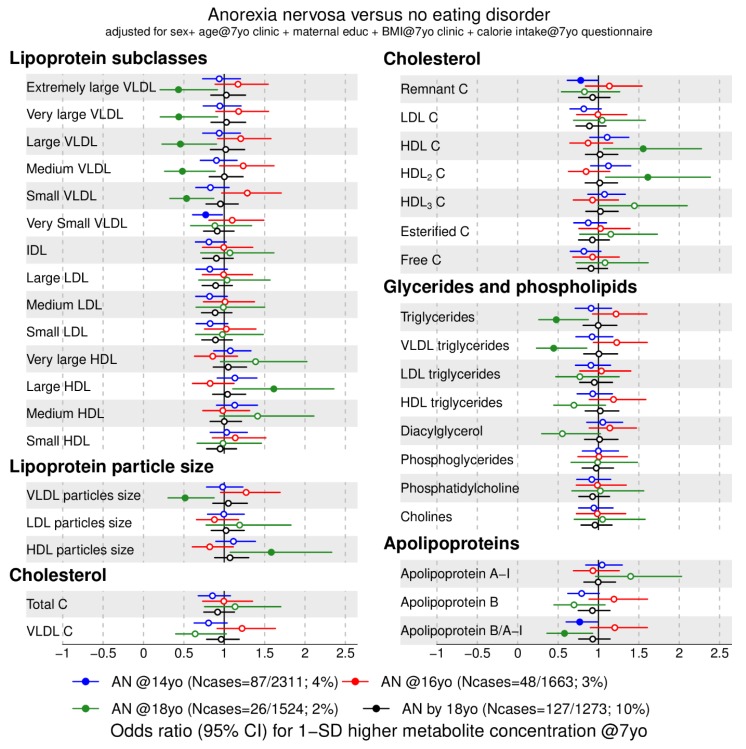
Estimated odds ratios for anorexia nervosa at 14, 16, 18 years of age and cumulatively across all three time points according to lipid-related metabolic trait concentrations at 7 years. Estimates refer to 1 standard deviation increase in metabolic trait concentration at 7 years (see also Appendix A). Error bars = 95% confidence intervals (CI). For lipoprotein subclasses, the total lipids (= triglycerides + phospholipids + total cholesterol) point estimate (and CIs) of each 14 subclasses is presented. Estimated odds ratios and corresponding 95% CIs for particle concentration and specific lipids in each lipoprotein subclass are given in Appendix A. Abbreviations: AN = anorexia nervosa; C = cholesterol; IDL = intermediate-density lipoprotein; LDL = low-density lipoprotein; HDL = high-density lipoprotein; VLDL = very low-density lipoprotein. Note: Filled dot: CI do not include the null.

**Figure 2 metabolites-09-00191-f002:**
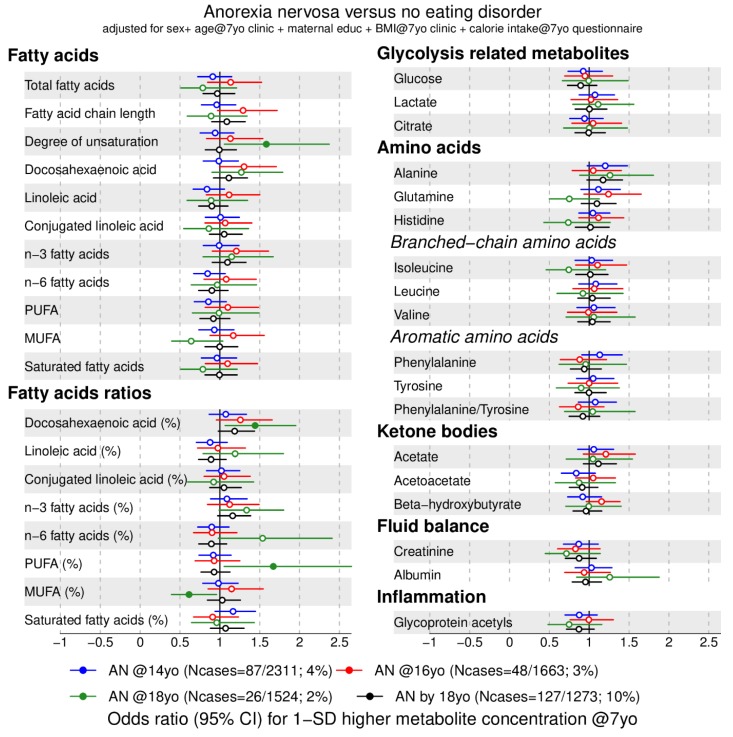
Estimated odds ratios for anorexia nervosa at 14, 16, 18 years of age and cumulatively across all three time points according to lipid- and non-lipid-related metabolic trait concentrations at 7 years (Figure 1 continued). Estimates refer to 1 standard deviation increase in metabolic trait concentration at 7 years. Error bars = 95% confidence intervals (CI). Abbreviations: AN = anorexia nervosa; MUFA = monounsaturated fatty acids; PUFA = polyunsaturated fatty acids. Note: Filled dot: CI do not include the null. MUFA, PUFA and saturated fatty acid concentrations include all fatty acids detected which have one, more than one, or zero C=C double bonds in their backbone, respectively.

**Figure 3 metabolites-09-00191-f003:**
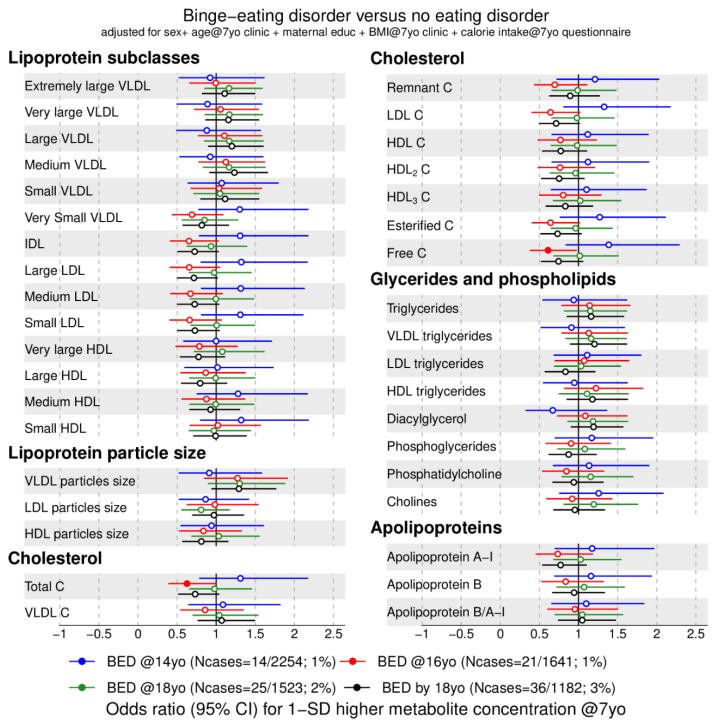
Estimated odds ratios for binge-eating disorder at 14, 16, 18 years of age and cumulatively across all three time points according to lipid-related metabolic trait concentrations at 7 years. Estimates refer to 1 standard deviation increase in metabolic trait concentration at 7 years (see also Appendix A). Error bars = 95% confidence intervals (CI). For lipoprotein subclasses, the total lipids (= triglycerides + phospholipids + total cholesterol) point estimate (and CIs) of each 14 subclasses is presented. Estimated odds ratios and corresponding 95% CIs for particle concentration and specific lipids in each lipoprotein subclass are given in Appendix A. Abbreviations: BED = binge-eating disorder; C = cholesterol; IDL = intermediate-density lipoprotein; LDL = low-density lipoprotein; HDL = high-density lipoprotein; VLDL = very low-density lipoprotein. Note: Filled dot: CI do not include the null.

**Figure 4 metabolites-09-00191-f004:**
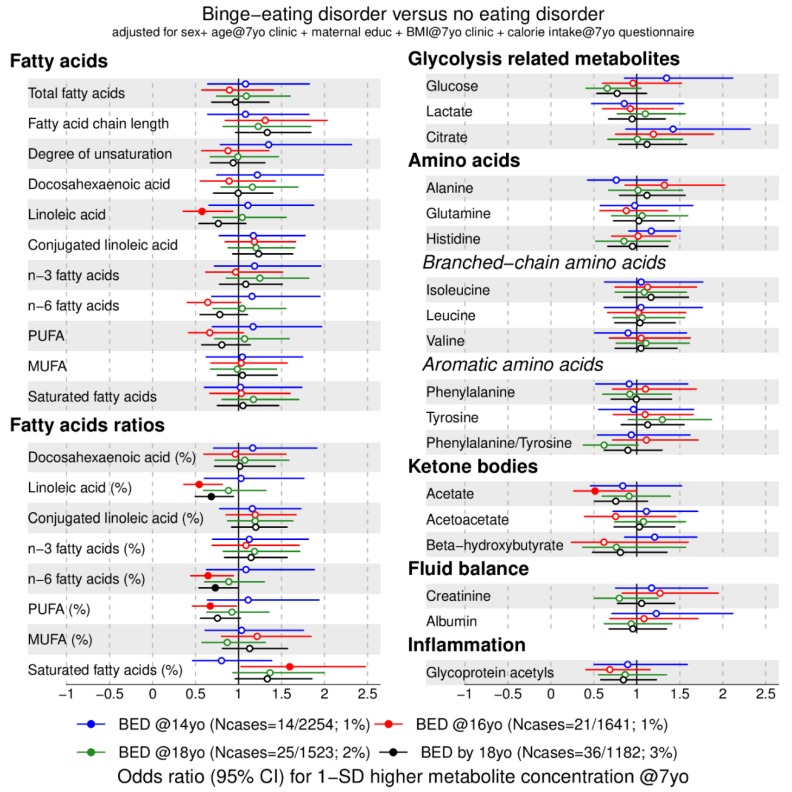
Estimated odds ratios for binge-eating disorder at 14, 16, 18 years of age and cumulatively across all three time points according to lipid- and non-lipid-related metabolic trait concentrations at 7 years (Figure 3 continued). Estimates refer to 1 standard deviation increase in metabolic trait concentration at 7 years. Error bars = 95% confidence intervals (CI). Abbreviations: BED = binge-eating disorder; MUFA = monounsaturated fatty acids; PUFA = polyunsaturated fatty acids. Note: Filled dot: CI do not include the null. MUFA, PUFA and saturated fatty acid concentrations include all fatty acids detected which have one, more than one, or zero C=C double bonds in their backbone, respectively.

**Table 1 metabolites-09-00191-t001:** Participant characteristics (*n* = 2929) at 7 years old stratified by eating disorders diagnosis at age 14, 16, 18 years old and cumulatively across all available time points (i.e., eating disorder diagnosis at age 14, 16, or 18 years old) in the Avon Longitudinal Study of Parents and Children (ALSPAC). Eating disorders diagnoses were derived using questionnaire data and DSM-5 criteria. Note: ***** indicates *p*-value < 0.006.

Eating Disorders Status by Assessment Age	*n* (%)	Female [*n* (%)]	Age (years) [mean (SD)]	Body Mass Index (kg/m^2^) [mean (SD)]	Average Total Calorie Intake (kcal) [mean (SD)]	Mother >12 Years in Education [*n* (%)]
Anorexia Nervosa (*N* = 2907)						
Diagnosis at 14 Years Old						
No Eating Disorder	2224 (96)	1057 (48)*	7.5 (0.1)	15.9 (1.8)*	1726.3 (304.7)*	1113 (50)
Anorexia Nervosa	87 (4)	61 (70)*	7.5 (0.1)	14.6 (1.3)*	1639.4 (255.8)*	40 (46)
Diagnosis at 16 Years Old						
No Eating Disorder	1615 (97)	729 (45)*	7.5 (0.1)	15.7 (1.6)*	1729.4 (305.0)*	853 (53)
Anorexia Nervosa	48 (3)	37 (77)*	7.4 (0.1)	14.5 (0.9)*	1565.8 (261.0)*	25 (52)
Diagosis at 18 Years Old						
No Eating Disorder	1498 (98)	697 (47)*	7.4 (0.1)	15.8 (1.7)*	1728.9 (304.1)	814 (54)
Anorexia Nervosa	26 (2)	21 (81)*	7.5 (0.1)	14.6 (1.4)*	1673.7 (248.9)	15 (58)
Cumulatively Across All Available Time Points (i.e., Eating Disorder Diagnosis at Age 14, 16, or 18 Years Old)						
No Eating Disorder	1146 (90)	655 (57)	7.4 (0.1)	16.1 (1.8)*	1725.9 (301.6)*	650 (57)
Anorexia Nervosa	127 (10)	88 (69)	7.5 (0.1)	14.6 (1.2)*	1636.8 (267.5)*	63 (50)
Binge-Eating Disorder (*N* = 2879)						
Diagnosis at 14 Years Old						
No Eating Disorder	2240 (99)	1070 (48)	7.5 (0.1)	15.9 (1.8)	1723.3 (304.6)	1122 (50)
Binge-Eating Disorder	14 (1)	8 (57)	7.4 (0.1)	16.4 (1.1)	1714.6 (422.2)	4 (29)
Diagnosis at 16 Years Old						
No Eating Disorder	1620 (99)	732 (45)*	7.5 (0.1)	15.7 (1.6)*	1729.7 (304.5)	857 (53)
Binge-Eating Disorder	21 (1)	17 (81)*	7.4 (0.2)	16.7 (1.4)*	1854.9 (263.8)	10 (48)
Diagnosis at 18 Years Old						
No Eating Disorder	1498 (98)	697 (47)*	7.4 (0.1)	15.8 (1.7)	1728.9 (304.1)	814 (54)
Binge-Eating Disorder	25 (2)	20 (80)*	7.4 (0.1)	16.8 (1.9)	1744.1 (329.6)	15 (60)
Cumulatively Across All Available Time Points (i.e., Eating Disorder Diagnosis at Age 14, 16, or 18 Years Old)						
No Eating Disorder	1146 (97)	655 (57)	7.4 (0.1)	16.1 (1.8)	1725.9 (301.6)	650 (57)
Binge-Eating Disorder	36 (3)	27 (75)	7.4 (0.1)	16.6 (1.7)	1808.7 (267.8)	21 (58)

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
