# Peer review of "Associations between Blood Metabolic Profile at 7 Years Old and Eating Disorders in Adolescence: Findings from the Avon Longitudinal Study of Parents and Children"

_metabolites, 2019, doi:10.3390/metabo9090191_

Round 1
Reviewer 1 Report
The study design was appropriate for this exploratory study. That being said, the samples used to infer eating disorders metabolic traits risk factors at age of 7 may not be representative of population-based subjects with EDs. The cumulative prevalence rates for AN and BN were 10% and 3% respectively. Aren't these rates very different from prior literature suggests? What does this say about the samples chosen? The lack of fasting state is actually more problematic than authors suggested. Also, the lack of accounting on dietary compositions at age 7 is likely to confound the findings. The metabolic traits at age 7 are significantly influenced by the genomic profile, which should also be described and considered. Finally, since the authors have access to the data of mothers, what generational factors can they identify to support their conclusion? This exploratory study contains a lot of data and the current presentation seems to be a bit on the simplistic side. Providing more details on the rationale, methods, and results as well as having a more comprehensive discussion on their findings are recommended.
Author Response
Bristol, 26 August 2019
Dear Editors,
We thank the reviewers for their generous comments on the manuscript and have edited the manuscript to address their concerns.
Below we have responded to each of the reviewers’ comments in turn and highlighted where changes have been made to the original manuscript, additionally we also submitted the track changed manuscript. We also removed most abbreviations throughout the manuscript to ease readership.
As per requested by the editor we have [1] used the word doc formatted by the editorial office, [2] have indicated the ethic code on subsection 4.1 Study population of the Materials and Methods section of the manuscript and [3] have changed the text highlighted by the similarity report.
We would like to request that in case the manuscript is accepted for publication that a note is added to indicate that authors Diana L. Santos Ferreira and Christopher Hübel contributed equally to the presented work.
We look forward to hearing from you and would be glad to respond to any further questions.
Yours Sincerely,
Christopher Hübel and Diana L. Santos Ferreira (on behalf of the other authors)
REVIEWER 1:
Q1.1: The study design was appropriate for this exploratory study. That being said, the samples used to infer eating disorders metabolic traits risk factors at age of 7 may not be representative of population-based subjects with EDs. The cumulative prevalence rates for AN and BN were 10% and 3% respectively. Aren't these rates very different from prior literature suggests? What does this say about the samples chosen?
RESPONSE: We thank the reviewer for pointing out this error, the percentages are not prevalences as we only used a subsample of the whole ALSPAC cohort. We have amended the sentence on page 3 and hope it is now clearer:
“Binge-eating was diagnosed in 1%, 1% and 2% of participants at 14, 16 and 18 years, respectively; and amongst 3% of participants by age 18 years (cumulatively at all three time points) (Table 1).”
Additionally, the diagnostic criteria for eating disorders have changed in the last three years with the advent of DSM-5 broaden the diagnosis of anorexia nervosa and introducing the diagnosis of binge-eating disorder for the first time. Newer studies in population samples with improved screening and better diagnostic procedure imply the occurrence of more cases than previously assumed (Micali, 2017). Moreover, only about 30-50% of patients with eating disorders seek treatment complicating the estimation of prevalence in the general population.
Micali, N., Martini, M. G., Thomas, J. J., Eddy, K. T., Kothari, R., Russell, E., … Treasure, J. (2017). Lifetime and 12-month prevalence of eating disorders amongst women in mid-life: a population-based study of diagnoses and risk factors. BMC Medicine, 15(1), 12.
Q1.2: The lack of fasting state is actually more problematic than authors suggested.
RESPONSE: We share the reviewer’s concern and have clearly stated in the Discussion section that it is a limitation of our work. However, it is important to underscore that the ALSPAC cohort represents the only longitudinal data set that has profiled metabolites and eating disorder phenotypes prospectively. We are lucky to be able to investigate such a rich data set whose metabolomic data at age 7 was collected around 2002.
Because of the concern shared with the reviewer, we extensively discuss how the non-fasting status may affect interpretation of our results. We have discussed (1) how fasting status could bias associations (on page 11, “If individuals who experience eating disorders in adolescence already exhibited eating disorders behaviors at age 7, then blood fasting status could confound associations for fasting-labile metabolic traits.”), (2) which metabolites associations are more likely to be biased if fasting-status is a confounder (“ (…) glucose and extremely large very-low density traits (…) may be affected as the latter have (…)”), (3) have cited the overall conclusion from studies that have explored how blood metabolome is affected by fasting status (“However, most blood metabolic traits seem to be unaffected by fasting status [23-27].”), (4) have stated how we adjusted for this potential confounder (“Nonetheless, we anticipate that most of this potential confounding would be captured by BMI and calorie intake measured at 7 years due to their association with the eating behaviors.”) and (5) have cautioned the reader regarding the interpretation of our results in the Discussion (“We, however, found no associations between anorexia nervosa case status in adolescence and glucose or glycolysis-related metabolites. However, this finding must be interpreted extremely cautiously as the participants had not fasted at age 7 when the blood samples were taken.”) and lastly have recommended that future research is done on fasting samples (“However, our results must be seen in the light of this limitation and future studies and data collection must sample participants after a fasting period.”).
Q1.3: Also, the lack of accounting on dietary compositions at age 7 is likely to confound the findings.
RESPONSE: Our models are adjusted for BMI and total calorie intake (both at 7 years) and both are associated with dietary composition at 7 years, therefore we expect that potential confounding by the latter would be adjusted in our models.
Q1.4: The metabolic traits at age 7 are significantly influenced by the genomic profile, which should also be described and considered.
RESPONSE: Thank you for this important point. We believe that the addition of genetic data is beyond the scope of our original analysis plan of our exploratory study, but we are planning on adding polygenic scores in follow-up investigations. Two large-scale genome-wide association studies of anorexia nervosa have interrogated the metabolic component of this illness (Watson et al., Nature Genetics, 2019) by showing that anorexia nervosa genetically correlates with insulin sensitivity and HDL cholesterol. This has given weight to the hypothesis explored in our study that metabolic alterations are ‘traits’ of eating disorders and therefore precede their onset. Therefore, in this hypothesis-generating study we set out to explore the prospective association between metabolic traits in childhood and eating disorders in adolescence. However, we are indirectly controlling for parts of the genomic profile in the adolescents by including phenotypic BMI, calorie intake, and maternal education, which all show a heritable component themselves, as covariates in our analyses. Additional inclusion of genetic profiles (e.g., polygenic scores) could also overcorrect that model as the metabolite concentration lies on the causal pathway from genetic liability over metabolite to eating disorder phenotype. This investigation would need a formal mediation analysis, which is out of the scope of our exploratory study.
Q1.5: Finally, since the authors have access to the data of mothers, what generational factors can they identify to support their conclusion?
RESPONSE: Our exploratory study investigated whether alterations in metabolites at age 7 are prospectively associated with a diagnosis of an eating disorder in adolescence to generate new hypotheses and identify metabolites as potential risk factors for eating disorder pathology. Investigating the maternal (genetic and environmental) effects is an interesting approach and planned for follow-up studies. For this study, we are indirectly controlling for parts of the maternal and child genomic profile by including phenotypic BMI, calorie intake, and maternal education, which all show a heritable component themselves, as covariates.
Q1.6: This exploratory study contains a lot of data and the current presentation seems to be a bit on the simplistic side. Providing more details on the rationale, methods, and results as well as having a more comprehensive discussion on their findings are recommended.
RESPONSE: Please see response to REVIEWER 2, Q2.1. In addition, we added the following changes to make it clearer to the reader the rationale, methods and discussion:
On the Introduction (page 2), we give now information on the statistical methods used:
“In this exploratory study, we assessed for the first time the prospective associations between blood metabolome at 7 years and AN and BED in adolescence, using data from the Avon Longitudinal Study of Parents and Children (ALSPAC), a large UK birth cohort. Logistic regression was used to examine the associations of each EDTA-plasma metabolic traits at 7 years old with either of AN and BED assessed at 14, 16, 18, and cumulatively by 18 years.”
On the Results (page 2):
We have restructured this section into: 2.1 Descriptives, 2.2. Metabolites associated with anorexia nervosa, 2.3 Metabolites associated with binge-eating disorder and 2.4 Sensitivity analysis (previously there was only two sections 2.1. Anorexia nervosa and 2.2 Binge-eating disorder). And have made substantial changes to the Results section text (please see track changed manuscript) to make it clearer to the reader. We also added the following text at the beginning of the Results section:
“We investigated the association between 158 metabolite traits at 7 years with anorexia nervosa (Figure 1-2) and binge-eating disorder (Figure 3-4) at 14 (blue), 16 (red), 18 years (green), and cumulatively until 18 years (black). For our main analyses, we fitted 1264 logistic regression models that indicate whether an elevated metabolite is associated with lower or higher risk of an eating disorder diagnosis in adolescence. We present the data by plotting the point estimates (circles or dots) and respective confidence intervals (error bars) in forest plots (Figure 1-4) where metabolic traits are clustered by biochemical classes defined by core chemical structure (e.g., fatty acids) or metabolic pathway (e.g., glycolysis). In Forest plots, the points to right of the vertical black line that marks the baseline risk (i.e., 1) indicate elevated risk for an eating disorder whereas points on the left of the black vertical line indicate lower risk for an eating disorder in adolescence.”
In addition, we have restructured the discussion section to make it easier to follow. We now start with a summary and contextualization of the anorexia nervosa findings, followed by a summary of the binge-eating disorder related results, finishing with a limitation section. In addition, we have removed all abbreviations and we hope that these changes make the discussion more comprehensive and accessible.
Reviewer 2 Report
The manuscript report a complex study on ED metabolic traits, tentatively revealing a possible association between 7 years babies blood metabolome and the occurence of ED at 14, 16 or 18 years. Although the huge amount of statistics, the results are difficult to follow and the format used to report results (Fig 1-4, and those reported in the Supplementary file) is heavy to read for non experts. In the Supplementary may be better to assign different numbers to Figures, i.e. instead of three Figs 1, may be better to use 1a, 1b, 1c. The conclusions are the same difficult to follow.
Author Response
Bristol, 26 August 2019
Dear Editors,
We thank the reviewers for their generous comments on the manuscript and have edited the manuscript to address their concerns.
Below we have responded to each of the reviewers’ comments in turn and highlighted where changes have been made to the original manuscript, additionally we also submitted the track changed manuscript. We also removed most abbreviations throughout the manuscript to ease readership.
As per requested by the editor we have [1] used the word doc formatted by the editorial office, [2] have indicated the ethic code on subsection 4.1 Study population of the Materials and Methods section of the manuscript and [3] have changed the text highlighted by the similarity report.
We would like to request that in case the manuscript is accepted for publication that a note is added to indicate that authors Diana L. Santos Ferreira and Christopher Hübel contributed equally to the presented work.
We look forward to hearing from you and would be glad to respond to any further questions.
Yours Sincerely,
Christopher Hübel and Diana L. Santos Ferreira (on behalf of the other authors)
REVIEWER 2:
The manuscript report a complex study on ED metabolic traits, tentatively revealing a possible association between 7 years babies blood metabolome and the occurence of ED at 14, 16 or 18 years.
Q2.1: Although the huge amount of statistics, the results are difficult to follow and the format used to report results (Fig 1-4, and those reported in the Supplementary file) is heavy to read for non-experts.
RESPONSE: We appreciate the reviewer’s comment that our work includes a lot of results. This is common in metabolomics literature as metabolomics platforms quantify hundreds of metabolites from a biological sample, which is two orders of magnitude more than standard clinical chemistry assays commonly used in epidemiology studies. The platform used in our study quantified 158 metabolic traits. In addition, we investigated the association between these 158 traits (at 7 years) with anorexia nervosa and binge-eating disorder at 14, 16 and 18 years and cumulatively until 18 years, therefore, only for our main analysis, we fitted 1264 logistic regression models. A common way to present such data rich metabolomic-epidemiology results is by plotting the point estimates and respective confidence intervals in forest plots where metabolic traits are clustered by biochemical classes (defined by core chemical structure, e.g. fatty acids; or metabolic pathway e.g. glycolysis). Forest plots are an intuitive and quick way to assess direction and magnitude of associations and spot patterns amongst metabolic traits associations. In addition, for simplicity, in Figure 1 and 3, underneath the lipoprotein subclasses header, we plotted the point estimate (and confidence intervals) for total lipids concentration of each of the 14 lipoprotein subclasses instead of all traits of each of these subclasses (more information regarding lipoprotein subclasses and respective traits is provided in Appendix A: Lipoprotein subclasses and fatty acids details). Readers are referred to the online Supplementary Material for results regarding particle concentration or specific lipids of each 14 lipoprotein subclasses (previous Figure S1 and S6, now Figure S1a-S1c and S6a-S6c, as per requested by this reviewer). To make this clear to the reader we have added the following on the caption of Figure 1 and 3:
“For lipoprotein subclasses, the total lipids (= triglycerides + phospholipids + total cholesterol) point estimate (and CIs) of each 14 subclasses is presented. Estimated odds ratios and corresponding 95% CIs for particle concentration and specific lipids in each lipoprotein subclass are given in (…)”
Alongside the forest plots, we also included, in the Supplementary File, tables presenting the point estimates, confidence intervals and p-values of each model for the interested reader.
There is an extensive body of literature that likewise uses both forest plots alongside tables to report metabolomic-epidemiology results, please see work of Peter Wurtz, Mika Ala-Korpela, Debbie Lawlor.
Additionally, we summarised the methods in a brief paragraph before presenting the results and restructured the results by adding introductory sentences:
“We investigated the association between 158 metabolite traits at 7 years with anorexia nervosa (Figure 1-2) and binge-eating disorder (Figure 3-4) at 14 (blue), 16 (red), 18 years (green), and cumulatively until 18 years (black). For our main analyses, we fitted 1264 logistic regression models that indicate whether an elevated metabolite is associated with lower or higher risk of an eating disorder diagnosis in adolescence. We present the data by plotting the point estimates (circles or dots) and respective confidence intervals (error bars) in forest plots (Figure 1-4) where metabolic traits are clustered by biochemical classes defined by core chemical structure (e.g., fatty acids) or metabolic pathway (e.g., glycolysis). In Forest plots, the points to right of the vertical black line that marks the baseline risk (i.e., 1) indicate elevated risk for an eating disorder whereas points on the left of the black vertical line indicate lower risk for an eating disorder in adolescence.”
Q2.2: In the Supplementary may be better to assign different numbers to Figures, i.e. instead of three Figs 1, may be better to use 1a, 1b, 1c.
RESPONSE: We thank the reviewer for this suggestion and have changed the supplementary figures numbering in the supplementary material and in the main manuscript, now the three Figures S1 are Figure S1a, S1b, S1c; the two Figures S2 are Figure S2a, S2b and so forth.
Q2.3: The conclusions are the same difficult to follow.
RESPONSE: Please see response to REVIEWER 1, Q1.6.
Round 2
Reviewer 1 Report
The authors had made significant effort in revising this manuscript. The revision is done well. There are a few more changes that are recommended as described below.
The use of “metabolome” for this paper seems misleading. While it is technically true that these “exposures” (risk factor?), which were taken from subjects’ records from a prior study, were measured using (1H) NMR metabolomics platform and could be classified under “metabolome”, a majority of the markers (and the most significant markers identified to be associated with EDs) are lipid markers (e.g., HDL, LDL) used routinely in the clinical lab for various medical conditions. Cholesterol is not generally thought of as metabolite. An example metabolite of cholesterol is a group called oxysterols which are 27-carbon derivatives of cholesterol, or by-products of cholesterol biosynthesis, which were not measured in this study. The metabolic traits this paper described are very important but are not what most readers would expect as best examples of “metabolome”. In the Abstract, instead of “We associated the blood metabolome at 7 years (exposure) with risk for anorexia nervosa and binge-eating disorder…”, perhaps something similar to “We *measured associations between metabolic markers* at 7 years with risk for….………” would be better. Also in the Abstract, please clarify by stating “*Elevated* Linoleic acid and n-6 fatty acid ratios were associated with lower odds…..” On page 3: “First, we present the lipid metabolites that when elevated at age 7 years …..very-low density lipoprotein (all subclasses except very small VLDL), very-low density lipoprotein particle size … total triglycerides …very-low density lipoprotein triglycerides ….and monounsaturated fatty acid ratio …”. Relating to point 1, LDL and HDL are lipoproteins as they are a combination of fat (lipid) and protein while triglycerides are a type of fat (lipid). Monounsaturated fatty acid is chemically classified as a fatty acid containing a single double bond. None of these “markers” are typically described as “lipid metabolite”. Please revise these style of descriptions throughout the rest of the manuscript (including subsection titles) to minimize confusion. For Figures 2 and 4 in the manuscript and related figures in the supplement, please describe in the legend what fatty acids are included as components in each of the following: “n-3 fatty acids”, “n-6 fatty acids” , “MUFA”, “PUFA”, and “saturated fatty acids”.Author Response
REVIEWER 1:
Q1.1: The use of “metabolome” for this paper seems misleading. While it is technically true that these “exposures” (risk factor?), which were taken from subjects’ records from a prior study, were measured using (1H) NMR metabolomics platform and could be classified under “metabolome”, a majority of the markers (and the most significant markers identified to be associated with EDs) are lipid markers (e.g., HDL, LDL) used routinely in the clinical lab for various medical conditions. Cholesterol is not generally thought of as metabolite. An example metabolite of cholesterol is a group called oxysterols which are 27-carbon derivatives of cholesterol, or by-products of cholesterol biosynthesis, which were not measured in this study. The metabolic traits this paper described are very important but are not what most readers would expect as best examples of “metabolome”.
RESPONSE: We have deleted the word “metabolome” and substitute it by “metabolic traits”, “metabolic markers” or “metabolic profile” throughout the manuscript and supplementary material including titles, subtitles and abstract.
Q1.2: In the Abstract, instead of “We associated the blood metabolome at 7 years (exposure) with risk for anorexia nervosa and binge-eating disorder…”, perhaps something similar to “We *measured associations between metabolic markers* at 7 years with risk for….………” would be better. Also in the Abstract, please clarify by stating “*Elevated* Linoleic acid and n-6 fatty acid ratios were associated with lower odds…..”
RESPONSE: We have made the suggested corrections to the abstract: “We associated blood metabolic markers at 7 years (exposure) with risk for anorexia nervosa and binge-eating disorder (outcomes) at 14, 16, 18 years using logistic regression adjusted for maternal education, child’s sex, age(…)” and “Elevated linoleic acid and n-6 fatty acid ratios were associated with lower odds (…)”.
Q1.3: On page 3: “First, we present the lipid metabolites that when elevated at age 7 years …..very-low density lipoprotein (all subclasses except very small VLDL), very-low density lipoprotein particle size … total triglycerides …very-low density lipoprotein triglycerides ….and monounsaturated fatty acid ratio …”. Relating to point 1, LDL and HDL are lipoproteins as they are a combination of fat (lipid) and protein while triglycerides are a type of fat (lipid). Monounsaturated fatty acid is chemically classified as a fatty acid containing a single double bond. None of these “markers” are typically described as “lipid metabolite”. Please revise these style of descriptions throughout the rest of the manuscript (including subsection titles) to minimize confusion.
RESPONSE: Please see reply to Q1.1. That particular sentence now reads: ”First, we present the lipid markers that when elevated at age 7 years were associated with lower risk for binge-eating disorder at age 16 years, (…)”.
Q1.4: For Figures 2 and 4 in the manuscript and related figures in the supplement, please describe in the legend what fatty acids are included as components in each of the following: “n-3 fatty acids”, “n-6 fatty acids” , “MUFA”, “PUFA”, and “saturated fatty acids”.
RESPONSE: We thank the reviewer for this important suggestion. The following sentence was added to the relevant figures’ captions in the main manuscript and supplementary material:
“MUFA, PUFA and saturated fatty acid concentrations include all fatty acids detected which have one, more than one, or zero C=C double bonds in their backbone, respectively.”